# Impact of a short online course on the accuracy of non-ophthalmic diabetic retinopathy graders in recognising glaucomatous optic nerves in Vietnam

Olusola Oluyinka Olawoye [1,2] Thu Huong Ha,[3] Ngoc Pham,[4] Lam Nguyen,[5] David Hunter Cherwek,[6] Kayode Raphael Fowobaje,[7] Craig Ross,[8] Michael Coote,[8] Ving Fai Chan,[1] Malik Kahook,[9] Tunde Peto [10] Augusto Azuara-Blanco [11] Nathan Congdon[12,13,14]

For numbered affiliations see end of article.

**Correspondence to**
Dr Nathan Congdon;
ncongdon1@gmail.com

## ABSTRACT

**Purpose** To test an online training course for non-ophthalmic diabetic retinopathy (DR) graders for recognition of glaucomatous optic nerves in Vietnam.

**Methods** This was an uncontrolled, experimental, before-and-after study in which 43 non-ophthalmic DR graders underwent baseline testing on a standard image set, completed a self-paced, online training course and were retested using the same photographs presented randomly. Twenty-nine local ophthalmologists completed the same test without the training course. DR graders then underwent additional one-to-one training by a glaucoma specialist and were retested. Test performance (% correct, compared with consensus grades from four fellowship-trained glaucoma experts), sensitivity, specificity, positive and negative predictive value, and area under the receiver operating (AUC) curve, were computed.

**Results** Mean age of DR graders (32.6±5.5 years) did not differ from ophthalmologists (32.3±7.3 years, p=0.13). Online training required a mean of 297.9 (SD 144.6) minutes. Graders' mean baseline score (33.3%±14.3%) improved significantly after training (55.8%±12.6%, p<0.001), and post-training score did not differ from ophthalmologists (58.7±15.4%, p=0.384). Although grader sensitivity reduced before [85.5% (95% CI 83.5% to 87.3%)] versus after [80.4% (78.3% to 82.4%)] training, specificity improved significantly [47.8 (44.9 to 50.7) vs 79.8 (77.3 to 82.0), p<0.001]. Grader AUC also improved after training [66.6 (64.9 to 68.3)] to [80.1 (78.5 to 81.6), p<0.001]. Additional one-to-one grader training by a glaucoma specialist did not further improve grader scores.

**Conclusion** Non-ophthalmic DR graders can be trained to recognise glaucoma using a short online course in this setting, with no additional benefit from more expensive one-to-one training. After 5-hour online training in recognising glaucomatous optic nerve head, scores of non-ophthalmic DR graders doubled, and did not differ from local ophthalmologists. Intensive one-to-one training did not further improve performance

## STRENGTHS AND LIMITATIONS OF THIS STUDY

⇒ Use of an approach to assess the impact of training which included both a before–after comparison and external validation against the performance of local ophthalmologists.

⇒ Both the course and the assessment test reflected input from clinical experts, specialists in image-based screening and those engaged professionally in capacity building for the recognition of glaucomatous optic nerves.

⇒ A number of widely used indices of screening accuracy were computed, allowing the identification of specific benefits of training in improving specificity.

⇒ The generalisability of our results to other cadres of trainees and other settings is uncertain.

⇒ Further research is needed to better understand the practicality of this training model within the context of a full glaucoma screening programme.

living with glaucoma has increased from 64.3 million in 2013 to 76.0 million in 2020,[3] and is projected to rise a further 74% by 2040 due to an ageing and growing global population. In 2013, Asia accounted for 60% of the world's total glaucoma cases, followed by Africa at 13%.[3] The number of persons blind due to glaucoma increased by 0.8 million (62%) between 1990 and 2010, with most of this rise occurring in low and middle-income countries (LMICs).[4] This trend is likely to continue without new and more effective screening strategies appropriate for low-resource settings.

Glaucoma often progresses unnoticed until central visual acuity is permanently compromised; therefore, early detection of the disease, before symptoms develop, is key in reducing the impact of visual loss in these patients. The rate of undiagnosed glaucoma is high, especially in LMICs, where

## INTRODUCTION

Glaucoma is the leading cause of irreversible blindness globally.[1 2] The number of persons

approximately 95% of those affected are not aware of having the disease.[5 6]

Screening of any disease is a form of secondary prevention, with the goal of preventing undesirable outcomes. In the context of glaucoma, screening is designed to promote diagnosis, and treatment if necessary, during the asymptomatic stage in order to prevent further vision loss and blindness. So far, glaucoma screening has not been found to be cost effective in high-income countries.[7 8]

Recently, however, a decision-analytic Markov model has been used to show that population screening for both primary open-angle glaucoma (POAG) and primary angle closure glaucoma (PACG) was cost effective in both rural and urban China.[9] John and Parikh[10] reported that community screening for glaucoma in India would prevent 2190 person years of blindness over a 10-year period. They concluded that community population screening might be cost effective if targeted at persons aged 40–69 years, and if implemented in urban areas.

Unlike screening programmes for glaucoma, diabetic retinopathy (DR) screening is well established in many high-income countries, and increasingly in LMICs as well. It would seem logical that screening for two major blinding eye diseases, glaucoma and DR, using the same equipment and personnel would be more cost effective. It may be possible to train non-ophthalmic graders who are currently screening for DR to recognise glaucoma using the colour fundus images, reducing the diagnostic burden on scarce, highly trained healthcare providers in low-resource settings.

The aim of the current study is to develop and test an online course to train nurses, technicians and non-ophthalmologist physicians working as DR graders in a non-governmental organisation (NGO) programme in Vietnam to recognise glaucomatous optic nerves from ophthalmic images. The pre-training and post-training performance of graders are assessed on a standard test set of optic nerve images drawn from a variety of sources, and compared with the performance of local ophthalmologists not undergoing the same training.

## METHODS

Participating DR graders and ophthalmologists in Vietnam all provided written informed consent before recruitment into the study. Approval for the study was obtained from the Ethical Review Board of the Queen's University Belfast United Kingdom (MHLS 20_98), and the Ethics Committee of the Hanoi Medical College in Vietnam (IRB-VN01.001/IRB 00003121/ FWA 00004148, Approval number 587). Participants gave informed consent to participate in the study before taking part.

Patient and public involvement statement: None.

This is an uncontrolled, experimental, before-and-after study in which 43 non-ophthalmic DR graders were trained to screen for glaucoma using optic nerve photos. The online training course was developed specifically for the study and focused on a standardised approach to the pattern recognition of glaucomatous damage to the optic nerve. The study flow was as follows:

► Non-ophthalmic graders in a non-governmental (NGO) DR programme (DR graders) were consented, enrolled and underwent testing on the standard image set at baseline.
► The DR graders then completed the self-paced, online training course. Participants were encouraged to attempt practice questions placed at the end of each module of the course, and those failing to answer these correctly were asked to review the course again.
► DR graders then repeated the test on the standard image set. Images were identical to those at baseline, but were presented in random order. DR grader test scores were compared with baseline and also to those of 29 local ophthalmologists taking part in the NGO DR programme, but who did not take the optic nerve grading course.
► Finally, to determine whether further improvement was practical, the non-graders each underwent approximately 20–30 min of individually focused, one-to-one training by a glaucoma specialist (LN), and were tested a third time on the standard image set.

## Participants

We enrolled 43 graders currently working in the Orbis and Fred Hollow Foundation (FHF) DR screening project in Vietnam, excluding any ophthalmologists or optometrists. These included nurses, technicians and non-ophthalmic physicians (table 1). Graders had a previous 3-day training course on grading DR. After the training, they were encouraged to continue practising on their own until they were proficient in taking photos and grading these photos for DR. These DR graders were then enrolled to participate in the current study. Twenty-nine ophthalmologists in Vietnam from 20 eye hospitals and general hospitals in cities such as Hanoi, Da Nang and Ho Chi Minh City taking part in the project were also recruited for comparison.

The DR graders had already been trained to grade images for DR through the Orbis and FHF DR projects. Participants were informed that if they chose not to participate in the study; their jobs would not be affected in any way.

## Test set of optic nerve images

A total of 60 digital photographic images were used, 33 from eyes with glaucoma and 27 from healthy eyes. Among these, 19 images were obtained from the Orbis Vietnam program, 11 from the population-based Northern Ireland Cohort for the Longitudinal Study of Ageing (NICOLA) study and 23 from the Glaucomatous Optic Neuropathy Evaluation (GONE, https://gone-project.com/newgone/) training website. All images were of gradable quality in the opinion of the standard panel of glaucoma specialists, though clarity was not always optimal in order to mimic real world conditions. Images from the GONE website were categorised into

**Table 1** Characteristics of DR graders and ophthalmologists participating in a study of glaucomatous optic nerve grading

| Characteristics | Ophthalmologists (n=29) | DR graders (n=43) | T-test/$\chi^2$ (p value) |
|---|---|---|---|
| Age (mean±SD) years | 32.6±5.5 | 32.3±7.3 | 0.13 (0.893) |
| <30 years | 11 (37.9%) | 21 (48.8%) | 0.83 (0.361) |
| ≥30 years | 18 (62.1%) | 22 (51.2%) | |
| Sex | | | 1.99 (0.159) |
| Female (%) | 17 (58.6%) | 32 (74.4%) | |
| Male (%) | 12 (41.4%) | 11 (25.6%) | |
| Job title | | | |
| Nurse (%) | 0 (0.0%) | 18 (41.9%) | |
| Ophthalmologist (%) | 29 (100.0%) | 0 (0.0%) | |
| Other physician (%) | 0 (0.0%) | 22 (51.2%) | |
| Technician (%) | 0 (0.0%) | 3 (6.9%) | |
| Work experience (mean±SD) years | 7.21±5.2 | 8.22±7.1 | 0.65 (0.516) |
| <5 (%) | 11 (37.9%) | 18 (41.9%) | 0.11 (0.739) |
| ≥5 (%) | 18 (62.1%) | 25 (58.1%) | |

*Endocrinologist (5); general practitioner (14); radiologist (2); internist (1).

difficult, intermediate and easy (table 1) on the basis of the prior performance of several thousand website users in characterising them as normal or glaucomatous.[11] The GONE images were graded into normal and abnormal by investigators (NC, AA-B, MC, OO, all fellowship-trained glaucoma experts, and TP, a medical retina specialist with a clinical and research focus on image-based screening for eye disease). The images from NICOLA and the Orbis Vietnam program were also sorted into normal and abnormal based on characteristic glaucomatous features of the images by the same group of investigators. When there was internal disagreement between the investigators, the image was replaced with another on which all five agreed.

Although optic nerve images are considered personal data, all images in the test set were fully anonymised and de-identified. Blanket permission for research use of images was obtained through the NICOLA study, Orbis Vietnam DR programme and the GONE website, all of which have received prior ethics approval.

### Online training course

The online training course was delivered on the Orbis Cybersight website,(cybersight.org) and consisted of three modules (217 Vietnamese-language PowerPoint slides with voice-over narration in Vietnamese) describing a standardised, step-by-step process of evaluating the healthy and glaucomatous optic nerve head. Module 1 provided a simple introduction to glaucoma. Module 2 introduced the normal optic nerve head and a systematic description of its features, with some reference to images depicting glaucomatous damage. Module 3 described a systematic method for assessing optic nerve head photos to screen for glaucomatous features using "the 5 Cs": Cup to disc ratio, colour of the neuroretinal rim, comparison between the optic nerve heads in a single patient's two eyes, contour of the nerve, and 'concerns' (this includes optic disc haemorrhage, and papilloedema, or optic nerve oedema with blurred nerve head margins). At the end of modules 2 and 3, trainees were provided 50 normal optic nerve images and 50 images of glaucomatous nerves for practice, all drawn from the same three sources as the test set (the Orbis Vietnam DR programme, the NICOLA study and the GONE website).

Optic nerve photos suggestive of other ocular diseases besides glaucoma were excluded from the course, with the exception of potentially fatal conditions affecting the optic nerve, including optic nerve oedema as above. All modules were independently assessed and validated by the five investigators listed above (AA-B, MC, TP, NC, OO) for content and appropriateness for these cadres of health workers.

### Evaluation with the test set

As above, the set consisted of 60 images, which were presented without stereo on the Orbis Cybersight website, and were each graded as 'refer' (likely glaucoma) or 'do not refer' (normal) by the test-taker. For images graded as 'refer', the correct reason had to be selected from a list. A score of five marks was given for a correct 'no refer' decision. For a correct 'refer' decision, the participant received a score of five marks, and an additional five marks for every correct reason given for the referral, ranging from one to six reasons. No marks were deducted for incorrect responses. Each of the 60 questions was worth between 5 and 30 marks (depending on the number of correct reasons for referral), and a total of 470 marks were available on the test. A percentage score was calculated for each grader as the number of marks received divided by 470.

Correct answers were not revealed to the graders. The questions were presented in a random order, which differed between pre-test and post-test, and also from one test-taker to the other. None of the optic nerve images used in the practice test were included in the quiz.

### Additional focused one-to-one training

In order to determine if additional, measurable performance improvements could be realised with further, more intensive training, the DR graders each underwent 20–30 min of one-to-one training in optic nerve recognition by a glaucoma specialist (LN) to better distinguish presence of glaucomatous optic nerve damage. The training focused on individual areas of deficiency identified by the postcourse training and was conducted online due to COVID-19-related restrictions on face-to-face gatherings in Vietnam at the time. All 43 (100%) DR graders completed the additional training, and all underwent a final round of testing on the standard image set.

### Sample size and statistical methods

The minimum sample size of 42 participants (graders) was calculated using the paired sample size formula described by Hulley[12] with a confidence level of 95% and statistical power of 80%. Using the statistics from the GONE website, we estimated that the mean for our study participants at baseline (pre-training) would be at the fifth percentile in grading performance, while at the end of the course (post-training), they would be at the 50th percentile. Thus, our study was powered to detect an improvement from a score of 29%–60% correct answers between baseline and post-training performance on the test set.

True positives (TP) were images correctly graded as glaucomatous optic neuropathy while true negatives (TN) were those correctly graded as 'no glaucoma'. False positive (FP) were images that did not have any features suggestive of glaucomatous optic neuropathy but were incorrectly graded as glaucomatous, while false negatives (FN) were images with glaucomatous features that were incorrectly graded as normal by the test-taker. Sensitivity was then calculated as TP divided by the sum of TP and FN, specificity as TN divided by the sum of TN and FP, positive predictive value (PPV) as TP divided by the sum of TP and FP, and negative predictive value as TN divided by the sum of TN and FN. Additionally, area under the receiver operating characteristic (AUC) curve and kappa statistic were computed, with their 95% CIs. The performance of DR graders after training was compared with that before training, and also to that of ophthalmologists who had not taken the course.

Since this was a study involving DR graders and ophthalmologists, we did not involve patients and public in the design or conduct or reporting or dissemination of the study

## RESULTS

In total, 43 DR graders participated in the training course and completed both pre-training and post-training tests, and the additional test after the one-to-one training, while 29 Vietnamese ophthalmologists underwent a single test and did not take part in the training. All DR graders undergoing training completed three rounds of testing.

The mean age (32.6±5.5 years) did not differ from that of the ophthalmologists (32.3±7.3 years, p=0.893, table 1). There was also no statistically significant difference between the age, sex distribution and work experience of DR graders versus ophthalmologists. The DR graders required a mean of 297.8 min (144.6), to complete the course according to measurements carried out automatically by the Orbis Cybersight website.

DR graders improved significantly between their mean pre-training per cent scores (33.3%±14.3%) and their post-training performance score ((55.8%±12.6%), p<0.001)) (table 2). The mean post-training score for DR graders did not differ significantly from that for local ophthalmologists (58.7±15.4%, p=0.384]. These findings, significant improvement between pre-training and post-training tests and the lack of a significant difference between post-training tests by DR graders and the performance of ophthalmologists, were consistent across optic nerve images from all sources (table 2).

With respect to other indices of grading accuracy (table 3), there was significant change between DR graders' pre-training [85.5% (83.5% to 87.3%)] and post-training sensitivity [80.4% (78.3% to 82.4%), p<0.001], while specificity improved significantly [47.8 (44.9 to 50.7) to 79.8 (77.3 to 82.0), p<0.001]. The positive and negative predictive values also improved significantly between pre-training and post-training tests, as did the area under the ROC curve [66.6 (64.9 to 68.3)] to 80.1 (78.5 to 81.6), p<0.001) (table 3). Compared with the 29 ophthalmologists, the 43 DR graders had worse sensitivity, better specificity, worse negative predictive value and worse kappa after training. The PPV and the area under the ROC curve (our prespecified indicator of overall grading accuracy) did not differ between post-training DR graders and ophthalmologists (table 3).

Table 4 shows the predictors of the percentage of correct responses among study participants. In the univariate analysis, age<30 years (p=0.005) and experience<5 years (p<0.001) predicted higher scores. However, in the multivariate analysis, only experience<5 years remained significant (p=0.006). The difference between post-training graders and local ophthalmologists remained non-significant when adjusting for other factors.

The test performance of DR graders (n=43) did not improve significantly after additional focused, one-to-one training, when compared with their performance after the online training (table 5). Their mean scores still did not differ from the ophthalmologists after this additional focused training. These patterns were the same for optic nerve images from all sources

**Table 2** Accuracy (% Correct vs Gold Standard)* in grading optic nerve images from different sources, comparing DR graders and ophthalmologists

| Source of images | Number of images | % Pretraining Score of graders (A) Mean±SD | % Post-training Score of graders (B) Mean±SD | P value (A vs B) | % Scores of ophthalmologists (C) mean±SD | P value (B vs C) |
|---|---|---|---|---|---|---|
| GONE website | | | | | | |
| Easy | 8 | 31.9±17.6 | 59.2±17.6 | <0.001 | 65.1±21.9 | 0.216 |
| Moderate | 8 | 25.6±13.3 | 44.2±19.4 | <0.001 | 48.1±17.8 | 0.392 |
| Hard | 7 | 22.7±15.9 | 47.7±22.5 | <0.001 | 46.2±21.8 | 0.780 |
| Orbis Vietnam DR programme | 19 | 47.0±22.7 | 70.3±12.0 | <0.001 | 70.3±18.2 | 0.991 |
| NICOLA population-based study in Northern Ireland | 11 | 31.4±20.0 | 58.7±16.9 | <0.001 | 55.2±22.1 | 0.443 |
| Other images illustrating specific findings | 4 | 31.0±19.8 | 51.9±24.1 | <0.001 | 57.5±23.4 | 0.337 |
| Total† | 60 | 33.3±14.3 | 55.8±12.6 | <0.001 | 58.7±15.4 | 0.384 |

*A panel of four fellowship-trained glaucoma specialist ophthalmologists and one medical retina specialist with a clinical and research focus on image-based screening for eye disease (AA-B, MC, TP, NC, OO).
†The remaining three images came from a variety of other sources.

## DISCUSSION

This uncontrolled, experimental, before-and-after study found a significant improvement in the baseline mean scores of non-ophthalmic DR graders after undergoing a 5-hour training course. The post-course scores of the DR graders were almost double the baseline scores, and did not differ significantly from the scores of general ophthalmologists in Vietnam. This suggests that it is possible to train non-ophthalmic DR graders to screen accurately for glaucoma using optic nerve photos using a short, self-paced online course. Additionally, more intensive one-to-one training by an expert did not result in significant further improvement of grading performance, suggesting encouragingly that results obtained with a convenient and scalable tool like our course approach is what may be readily achievable with such cadres.

In many developing countries, half or more of patients with glaucoma present with advanced disease.[13–16] Shortage of providers limits access to eye care and leads to delays in diagnosis and management of glaucoma until late in the disease process, often too late for effective, sight-saving treatment.[17 18] For example, a recent study reported >60% prevalence of blindness in the operative eye among patients at the time of first glaucoma surgery in rural China.[19] Training non-ophthalmic graders to screen for glaucoma may be an initial step in promoting earlier glaucoma diagnosis in low-resource settings. This has been described as task shifting, a model in which lower cadres of staff are trained to do simpler tasks so that scarcer and more highly-trained workers such as ophthalmologists can concentrate on high-level medical decision-making and surgical management.[20–23]

The graders participating in this study were already trained to detect DR, the most frequent microvascular complication of diabetes and the leading cause of blindness in working-age adults. Studies[24–26] have demonstrated that >90% of severe vision loss from DR can be prevented through regular screening and prompt referral and treatment. Diabetic screening is more cost-effective compared with no screening or opportunistic screening.[27 28] Glaucoma is the most common cause of irreversible blindness globally[2–29] and glaucoma blindness can similarly be prevented by regular screening of high-risk persons, prompt referral and treatment. Training the same cadres to screen for these two major causes of blindness may further improve the cost-effectiveness and cost-benefit of screening. This model is of particular interest in low-income settings, where traditional eye health services are mainly located in urban centres, often with poor accessibility to rural populations.

The model assessed in the current study differs from that described in previous reports in several ways. Non-expert health workers were trained to recognise glaucoma using an online course, and their performance was objectively measured and compared with their baseline, and to the scores of local ophthalmologists who had not taken the course. Previous studies have used tele-medicine to send retinal images to reading centres for expert grading.[30–32] This can be challenging to sustain in low-resource settings, as it still depends on the time and availability of ophthalmologists or other relatively scarce trained experts to review retinal images for evidence of disease. Reading centres can also be logistically complex and expensive to establish and maintain in low resource

**Table 3** Accuracy in grading optic nerve images, comparing DR graders and ophthalmologists, and calculated against a panel of specialist ophthalmologists (the 'gold standard')

| Participant type | Sensitivity (95% CI) | Specificity (95% CI) | Positive predictive value (95% CI) | Negative predictive value (95% CI) | Kappa (95% CI) | Area under the receiver operator curve (95% CI) |
|---|---|---|---|---|---|---|
| DR graders, pretraining (%) | 85.5 (83.5 to 87.3) | 47.8 (44.9 to 50.7) | 66.7 (65.4 to 68.0) | 72.9 (70.1 to 75.6) | 34.4 (30.9 to 37.9) | 66.6 (64.9 to 68.3) |
| DR graders, post training(%) | 80.4 (78.3 to 82.4) | 79.8 (77.3 to 82.0) | 82.9 (81.2 to 84.5) | 76.9 (74.9 to 78.8) | 60.0 (56.9 to 63.1) | 80.1 (78.5 to 81.6) |
| P value comparing graders pretraining and post training | <0.001 | <0.001 | <0.001 | <0.011 | <0.001 | <0.001 |
| Ophthalmologists (no training given) (%) | 92.9 (91.1 to 94.4) | 73.8 (70.6 to 76.9) | 81.3 (79.4 to 83.0) | 89.5 (87.1 to 91.5) | 67.8 (64.3 to 71.3) | 83.4 (81.6 to 85.1) |
| P value comparing post-training graders and ophthalmologists | <0.001 | 0.002 | >0.05 | <0.05 | <0.05 | >0.05 |

DR, diabetic retinopathy.

settings, due to challenges such as the low bandwidth of available internet connections.

An advantage of training non-experts as in our current model is that they are more likely to be able to review the images in real time due to lack of competing clinical duties. In delayed grading models, such as those using telemedicine, patients have to be contacted later to notify them of their screening results. Existing studies in LMICs have shown that this can lead to substantial loss to follow-up.[33] The Rwanda Artificial Intelligence for Diabetic Retinopathy Screening (RAIDERS) study reported that participants receiving immediate feedback on the need for referral demonstrated higher adherence with recommended services compared with participants informed of the results of their screening several days later.[33]

Training with our course significantly improved graders' specificity and PPV. This effect of training is particularly relevant in low-resource settings, where the priority must be to avoid over-burdening already weak health systems with large numbers of false positive referrals. Ausayakhun et al[34] described a somewhat comparable model to ours, of training non-ophthalmologists to deliver point-of-service screening for glaucoma, cataract and age-related macular degeneration, and their reported specificity for glaucoma was approximately 80% using fundus photographs, similar to that observed in the current study.

Although the sensitivity of DR graders decreased after training, their specificity rose substantially. In screening programmes for a relatively rare disease such as glaucoma, affecting roughly 2%–4% of the population in most studies, avoiding false positives (good specificity) is most important for the cost-effectiveness of the programme. For example, assuming a 3% prevalence of glaucoma, the increase in specificity from 47.8% to 79.8% observed after training would reduce the number of false-positive referrals from 506 to 196 (61.3% more reduction) in a hypothetical population of 1000 persons, while the observed decline of sensitivity would result in only two more false negative cases. Although this is still a substantial number of false positives, it is fewer than the ophthalmologist would get, as the specificity of the graders post testing exceeds that of the ophthalmologists (79.8% vs 73.8%).

An important determinant of the PPV of a screening programme is the burden of disease in the target population.[35] The heavy burden of glaucoma among Asians[36–38] and Africans[39–41] has been well documented. It is estimated that Asia has the highest number of persons living with glaucoma globally, and that this figure will increase by 79.8% (POAG) and 58.4% (PACG) by 2040.[3] In view of this high glaucoma burden in Asia, and the growing gap between need and supply for ophthalmologists globally,[42] it is clear that new screening models are needed which rely on non-experts. We were able to train diabetic graders to perform at the level of local ophthalmologists, suggesting that our training course can deliver successful and effective task shifting.

**Table 4** Predictors of the percentage of correct responses on the test set among study participants

| Potential predictor | Univariate analysis | | | Multivariate analysis | | |
|---|---|---|---|---|---|---|
| | Beta | 95% CI | P value | Beta | 95% CI | P value |
| Age<30 years | 9.0 | 2.8 to 15.2 | 0.005 | 0.4 | -8.7 to 9.4 | 0.938 |
| Male sex | 0.4 | -6.5 to 7.4 | 0.899 | 1.8 | -4.7 to 8.3 | 0.585 |
| Post-training non-ophthalmic grader (vs ophthalmologist) | 2.9 | -9.5 to 3.7 | 0.384 | 3.1 | -9.2 to 3.1 | 0.319 |
| <5 years of work experience | 12.5 | 6.6 to 18.4 | <0.001 | 13.1 | 3.9 to 22.3 | 0.006 |
| Nurse (vs 'Other physicians', non-ophthalmic grader group only) | 4.5 | 12.3 to 3.3 | 0.251 | | | |
| Pretraining versus post training (DR graders only) | 0.4 | 0.1 to 0.6 | 0.007 | | | |

An important predictor of correct responses on the test set among study participants was working experience of<5 years. Lack of a positive effect of cumulative experience may seem counterintuitive, but we suggest this may reflect better performance among younger and more-recently trained cadres. Further studies are needed to replicate and better understand this finding.

Although artificial intelligence (AI) has been used successfully for DR screening,[43 44] well-trained human graders are crucial in the validation of these AI systems, which must employ locally derived images and personnel to be most reliable. Moreover, there are substantial existing legal and financial barriers to widespread sustainable use of AI for DR and glaucoma screening in many LMICs. Human systems are needed until these are overcome.

Strengths of this study include an approach to assessing the impact of training which included both a before–after comparison and external validation against the performance of local ophthalmologists. Success of the course against both of these criteria was consistent across images of varying difficulty and drawn from a wide variety of sources. Both the course and the assessment test reflected input from clinical experts, specialists in image-based screening and those engaged professionally in capacity building for the recognition of glaucomatous optic nerves. A number of widely used indices of screening accuracy were computed, allowing us to elucidate specific benefits of training in improving specificity.

This study has limitations as well. The generalisability of our results to other cadres of trainees and other settings is uncertain. Specifically, these results may not be directly applicable to novice graders without prior experience in interpreting fundus images.

Most importantly, further research is needed to better understand the practicality of this training model within the context of a full glaucoma screening programme. Additional work is under way to test the model as part

**Table 5** Accuracy (% Correct vs Gold Standard*) in grading optic nerve images from different sources: the effect of additional one-one training for DR graders after the online course

| Source of Images | Number of images | % Post-training score of graders after online course Mean±SD (A) (n=43) | % Post-training score of graders after additional focused one-one training, Mean±SD (B) (n=43) | P value (A vs B) | % Scores of ophthalmologists Mean±SD (C) (n=29) | P value (B vs C) |
|---|---|---|---|---|---|---|
| GONE website | | | | | | |
| Easy | 8 | 59.2±17.6 | 60.0±21.6 | 0.9364 | 65.1±21.9 | 0.6463 |
| Moderate | 8 | 44.2±19.4 | 48.7±21.9 | 0.6702 | 48.1±17.8 | 0.9529 |
| Hard | 7 | 47.7±22.5 | 49.7±25.8 | 0.8797 | 46.2±21.8 | 0.7886 |
| Orbis Vietnam DR programme | 19 | 70.3±12.0 | 69.8±14.8 | 0.9096 | 70.3±18.2 | 0.9265 |
| NICOLA population-based study in Northern Ireland | 11 | 58.7±16.9 | 66.7±14.8 | 0.2514 | 55.2±22.1 | 0.1670 |
| Other images illustrating specific findings | 4 | 51.9±24.1 | 56.1±26.4 | 0.8220 | 57.5±23.4 | 0.9393 |
| **Total | 60 | 55.8±12.6 | 60.3±16.4 | 0.1573 | 58.7±15.4 | 0.6422 |

of delivering outreach screening for glaucoma in sub-Saharan Africa, aimed at a broad variety of patients at risk for glaucoma, and not solely those with DR.

In summary, our study shows that it is possible to train non-ophthalmic DR graders quickly and at low cost to screen for glaucoma using optic nerve images. This model may improve glaucoma detection and ultimately reduce the prevalence of blindness from glaucoma in low-resource settings.

**Author affiliations**
[1]School of Medicine, Dentistry and Biomedical Sciences, Queens University Belfast, Belfast, UK
[2]Department of Ophthalmology, College of Medicine, University of Ibadan, Ibadan, Nigeria
[3]ORBIS International, Viet Nam, Viet Nam
[4]ORBIS International, New York, New York, USA
[5]Hanoi Medical University, Hanoi, Viet Nam
[6]Global Programs, ORBIS International, New York, New York, USA
[7]Department of Epidemiology and Medical Statistics, University of Ibadan, Ibadan, Nigeria
[8]Centre for Eye Research Australia, East Melbourne, Victoria, Australia
[9]University of Colorado at Colorado Springs, Colorado, UK
[10]Faculty of Medicine Health and Life Sciences, Queen's University Belfast, Belfast, UK
[11]Centre for Public Health, Queens University Belfast, Belfast, UK
[12]Department of Ophthalmology and Public Health, Queen's University Belfast, Belfast, UK
[13]Orbis International NY USA, New York, New York, USA
[14]Department of Ophthalmology, Zhongshan Ophthalmic Centre, Guangzhou, People's Republic of China

**Contributors** NC, OOO, AA-B, VFC, DHC and NP conceived and designed the study. All authors made inputs into the final design of the study. OOO drafted the initial course and the test for the graders and ophthalmologists. NC, AA-B, MK, MC, OOO, VFC, DHC, LN and TP revised the course and test contents and ensured standardisation for the study. OOO, KRF, NC, CR and THH conducted the analysis. OOO drafted the initial manuscript. NC, AA-B, MK, DHC, TP, MC, NP and VFC made major contributions to the drafting of the manuscript. All authors reviewed it critically for important intellectual content, and approved the final version to be published.NC is the senior author on the paper

**Funding** eXcellence in Ophthalmology and Vision Awards (XOVA Novartis), and Wellcome Trust grant number: 222490/Z/21/Z. Award grant Number: XOVA Novartis: N/A, Wellcome Trust: 222490/Z/21/Z.

**Competing interests** Congdon serves as a paid consultant for Belkin Vision, a company selling devices for the treatment of glaucoma. Kahook has an equity interest in SpyGlass Pharma and receives royalties from New World Medical and Alcon. Olusola Olawoye is a Consortium for Advanced Research and Training in Africa (CARTA) Fellow.

**Patient and public involvement** Patients and/or the public were not involved in the design, or conduct, or reporting, or dissemination plans of this research.

**Patient consent for publication** Not applicable.

**Ethics approval** Approval for the study was obtained from the Ethical Review Board of the Queen's University Belfast United Kingdom (MHLS 20_98), and the Ethics Committee of the Hanoi Medical College in Vietnam (IRB-VN01.001/IRB 00003121/ FWA 00004148, Approval number 587). Participants gave informed consent to participate in the study before taking part.

**Provenance and peer review** Not commissioned; externally peer reviewed.

**Data availability statement** Data are available upon reasonable request.

**ORCID iDs**
Olusola Oluyinka Olawoye http://orcid.org/0000-0003-2357-8924
Tunde Peto http://orcid.org/0000-0001-6265-0381
Augusto Azuara-Blanco http://orcid.org/0000-0002-4805-9322

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
