## [Reviewer comments · BMJ Open]

ARTICLE DETAILS

TITLE (PROVISIONAL)	Impact of a short online course on the accuracy of non-ophthalmic diabetic retinopathy graders in recognizing glaucomatous optic nerves in Vietnam
AUTHORS	Olawoye, Olusola; Ha, Thu Huong; Pham, Ngoc; Nguyen, Lam; Cherwek, David; Fowobaje, Kayode Raphael; Ross, Craig; Coote, Michael; Chan, Ving Fai; Kahook, Malik; Peto, Tunde; Azuara-Blanco, Augusto; Congdon, Nathan

VERSION 1 – REVIEW

REVIEWER	Nemeth, Janos Semmelweis Egyetem, Dept. of Ophthalmology
REVIEW RETURNED	06-Jul-2023

GENERAL COMMENTS	This is an excellent and important study with useful practical implications. The study design and protocol are well described and clear. There are some minor inaccuracies in the presentation of the results. The conclusions are based on the findings and justified. I support the publication after minor revision. Detailed comments: Page 14, Lines 21-23. It would be advisable to present the stated „non-significant p value” also for the change in sensitivity values before and after the training. Page 14 Line 21-25 and Table 3. Please, note and correct that the presentation of significance is different in the text and in Table 3 for DR graders pre- and post-training: e.g. for sensitivity: non-significant in the text and $p < 0.05$ decrease in Table 3.; specificity: < 0.001 vs < 0.05, respectively. Table 3. The sensitivity values of DR graders decreased after the training and was significantly lower than that of the ophthalmologist value. If this is the situation, need to be addressed this in the discussion although I agree with the authors that improvement in specificity and positive predictive value are more important in glaucoma screening.
--

REVIEWER	Rani, Padmaja Kumari Univ Alabama Birmingham
REVIEW RETURNED	12-Jul-2023

GENERAL COMMENTS	The reduction in sensitivity of graders after training and its implications is an important topic to address in the study. It is crucial to clarify why the sensitivity of the graders decreased after their training. Understanding the reasons behind this reduction can help identify potential challenges or limitations in the training program and inform improvements for future implementations. One implication of reduced sensitivity is that the trained graders may be
---

	more likely to miss cases of glaucoma during the screening process. This could lead to a higher rate of false negatives, where individuals with glaucoma are wrongly classified as not having the condition. Consequently, such false negatives could result in delayed or missed diagnoses, leading to potential vision loss or progression of the disease. Please clarify. To ensure the accuracy and quality of glaucoma screening programs, it is important to incorporate audit mechanisms. These mechanisms act as checks and balances to monitor the performance of the graders and the overall screening process. Regarding the study's graders, if they are already practicing diabetic retinopathy (DR) graders, it is important to highlight that the study results mainly apply to experienced graders with prior knowledge of fundus images. The limitations of the study should include the fact that these results may not be directly applicable to novice graders without any prior experience in interpreting fundus images. Additionally, it is crucial to provide details about the duration and nature of the DR training these graders received before participating in the glaucoma screening program. These details are essential to assess the replicability and applicability of similar training programs in other settings. Understanding the length and intensity of their previous training can help determine the level of expertise and transferability of their skills to the glaucoma screening context.
--	--

VERSION 1 – AUTHOR RESPONSE

Reviewer 1

1. Comments: This is an excellent and important study with useful practical implications. The study design and protocol are well-described and clear. There are some minor inaccuracies in the presentation of the results. The conclusions are based on the findings and are justified. I support the publication after minor revision.

Response: Thank you very much

2. Comments: Lines 21-23. It would be advisable to present the stated „non-significant p value” also for the change in sensitivity values before and after the training.

Response: This has been done both in the table (Table 3, page 20) and in the results (page 11)

3. Comments: Page 14 Line 21-25 and Table 3. Please, note and correct that the presentation of significance is different in the text and in Table 3 for DR graders pre- and post-training: e.g. for sensitivity: non-significant in the text and $p < 0.05$ decrease in Table 3.; specificity: < 0.001 vs < 0.05 , respectively.

Response: The exact p-value has been written on the table This has been corrected (Table 3, page 20) and page 11 under results

4. Comments: The sensitivity values of DR graders decreased after the training and was significantly lower than that of the ophthalmologist value. If this is the situation, need to be addressed this in the discussion although I agree with the authors that improvement in specificity and positive predictive value are more important in glaucoma screening.

Response: Thank you for your comment. We have now added to the Discussion the following material: ““Although the sensitivity of DR graders decreased slightly after training, their specificity rose substantially. In screening programs for a relatively rare disease such as glaucoma, affecting roughly 2-4% of the population in most studies, avoiding false positives (good specificity) is most important for the cost-effectiveness of the program. For example, assuming a 3% prevalence of glaucoma, the increase in specificity from 47.8% to 79.8% observed after training would reduce the number of false positive referrals from 506 to 196 (61.3% more reduction) in a hypothetical population of 1000 persons, while the observed decline of sensitivity would result in only two more false negative cases.”

Although this is still a substantial number of false positives, it is fewer than the ophthalmologist would get, as the specificity of the graders post testing exceeds that of the ophthalmologists (79.8 vs 73.8%). Page 14 second paragraph under discussion

Reviewer 2

Comments: The reduction in sensitivity of graders after training and its implications is an important topic to address in the study. It is crucial to clarify why the sensitivity of the graders decreased after their training. Understanding the reasons behind this reduction can help identify potential challenges or limitations in the training program and inform improvements for future implementations. One implication of reduced sensitivity is that the trained graders may be more likely to miss cases of glaucoma during the screening process. This could lead to a higher rate of false negatives, where individuals with glaucoma are wrongly classified as not having the condition. Consequently, such false negatives could result in delayed or missed diagnoses, leading to potential vision loss or progression of the disease. Please clarify. To ensure the accuracy and quality of glaucoma screening programs, it is important to incorporate audit mechanisms. These mechanisms act as checks and balances to monitor the performance of the graders and the overall screening process

Response: Thank you for your comment. We have now added to the Discussion the following material: ““While the sensitivity of DR graders decreased slightly after training, their specificity rose substantially. In screening programs for a relatively rare disease such as glaucoma, affecting roughly 2-4% of the population in most studies, avoiding false positives (good specificity) is most important for the cost-effectiveness of the program. (Page 15 paragraph one under discussion). For example, assuming a 3% prevalence of glaucoma, the increase in specificity from 47.8% to 79.8% observed after training would reduce the number of false positive referrals from 506 to 196 (a 61.3% decrease) in a hypothetical population of 1000 persons, while the observed decline of sensitivity would result in only two more false negative cases.” Although this is still a substantial number of false positives, it is fewer than the ophthalmologist would get, as the specificity of the graders post testing exceeds that of the ophthalmologists (79.8 vs 73.8%). Page 14 second paragraph under discussion

Comments: Regarding the study's graders, if they are already practicing diabetic retinopathy (DR) graders, it is important to highlight that the study results mainly apply to experienced graders with prior knowledge of fundus images. The limitations of the study should include the fact that these results may not be directly applicable to novice graders without any prior experience in interpreting fundus images.

Response: We had already included the following statement in our limitations section in our initial submission: The generalizability of our results to other cadres of trainees and other settings is uncertain Paragraph 1 page 16 under discussion

However, to further clarify this we have added the below statement: Specifically, these results may not be directly applicable to novice graders without prior experience in interpreting fundus images. Page 16 paragraph 1 under discussion.

Reviewers comments: Additionally, it is crucial to provide details about the duration and nature of the DR training these graders received before participating in the glaucoma screening program. These details are essential to assess the replicability and applicability of similar training programs in other settings. Understanding the length and intensity of their previous training can help determine the level of expertise and transferability of their skills to the glaucoma screening context.

Response: Thank you very much for the comment. To address this, we have added the following statement on page 6 under participant lines 3-5

Graders had a previous 3-day training course on grading diabetic retinopathy (DR). After the training, they were encouraged to continue practicing on their own until they were proficient in taking photos and grading these photos for DR.

VERSION 2 – REVIEW

REVIEWER	Rani, Padmaja Kumari Univ Alabama Birmingham
REVIEW RETURNED	02-Sep-2023
GENERAL COMMENTS	All comments have been addressed.